# EBV-Associated Hub Genes as Potential Biomarkers for Predicting the Prognosis of Nasopharyngeal Carcinoma

**DOI:** 10.3390/v15091915

**Published:** 2023-09-12

**Authors:** Tengteng Ding, Yuanbin Zhang, Zhixuan Ren, Ying Cong, Jingyi Long, Manli Peng, Oluwasijibomi Damola Faleti, Yinggui Yang, Xin Li, Xiaoming Lyu

**Affiliations:** 1Shenzhen Key Laboratory of Viral Oncology, The Clinical Innovation & Research Centre (CIRC), Shenzhen Hospital of Southern Medical University, Shenzhen 518100, China; dhyzf1214@163.com (T.D.); laozhang859968@163.com (Y.Z.); egocong@126.com (Y.C.); pengmanlinfykdx@ouklook.com (M.P.); 2The Third School of Clinical Medicine, Southern Medical University, Guangzhou 510630, China; ljyw61@163.com (J.L.); faletisijibomi@gmail.com (O.D.F.); 3Department of Radiation Oncology, Huadong Hospital, Fudan University, Shanghai 200040, China; rzx806410@163.com; 4Department of Laboratory Medicine, The Third Affiliated Hospital, Southern Medical University, Guangzhou 510630, China; 5Department of Urology, Shenzhen Hospital of Southern Medical University, Shenzhen 518100, China

**Keywords:** Epstein–Barr virus, nasopharyngeal carcinoma, prognosis, hub genes

## Abstract

This study aimed to develop a model using Epstein–Barr virus (EBV)-associated hub genes in order to predict the prognosis of nasopharyngeal carcinoma (NPC). Differential expression analysis, univariate regression analysis, and machine learning were performed in three microarray datasets (GSE2371, GSE12452, and GSE102349) collected from the GEO database. Three hundred and sixty-six EBV-DEGs were identified, 25 of which were found to be significantly associated with NPC prognosis. These 25 genes were used to classify NPC into two subtypes, and six genes (C16orf54, CD27, CD53, CRIP1, RARRES3, and TBC1D10C) were found to be hub genes in NPC related to immune infiltration and cell cycle regulation. It was shown that these genes could be used to predict the prognosis of NPC, with functions related to tumor proliferation and immune infiltration, making them potential therapeutic targets. The findings of this study could aid in the development of screening and prognostic methods for NPC based on EBV-related features.

## 1. Introduction

Nasopharyngeal carcinoma (NPC) is an epithelial carcinoma that originates from the nasopharyngeal epithelium [1]. NPC shows stark imbalances in geographical distribution, with a high incidence in southern China and Southeast Asia [2]. NPC can occur at any age, including in children, with a slow increase in incidence throughout life [3]. In 2020, it was estimated that around 133,354 individuals worldwide received a diagnosis of NPC and 80,008 died from this disease [4]. One of the distinctive features of NPC is its insidious onset in the nasopharynx (NP), making early symptoms difficult to detect, and so it usually reaches an intermediate or advanced stage before diagnosis [5,6]. NPC is also known for its aggressive and invasive nature, with a strong tendency to metastasize [6].

The prognosis of NPC varies depending on the stage of the disease. It is estimated that localized NPC has a 5-year relative survival rate of 82%, while regional spread reduces the rate to 72% and distant metastasis significantly lowers the rate to 49% [4]. Radiotherapy and chemotherapy are highly effective and can yield favorable outcomes in nonmetastatic NPC [7]. However, for distant metastatic nasopharyngeal carcinoma, recurrence, metastasis, and chemotherapy resistance remain significant challenges, despite the recent improvement in mortality rate [7,8]. The early prediction of incidence and poor prognosis is therefore crucial for NPC patients.

Epstein–Barr virus (EBV) infection is closely associated with the development of NPC, especially the nonkeratinizing type, as nearly 100% of these tumors show evidence of EBV infection [9]. EBV appears to have a broad impact on tumor development through its ability to induce genetic changes that affect various cellular processes, such as cell proliferation, migration, invasion, epithelial–mesenchymal transition (EMT), and angiogenesis [10,11,12,13,14,15]. EBV even alters the immune infiltration of tumors [16], further emphasizing the broad and complex effects of EBV on cancer biology.

The development of screening and prognostic methods for nasopharyngeal carcinoma based on EBV-related features is therefore an attractive research area. Conventional prediction or risk stratification models are mainly focused on clinical features, such as serum EBV DNA markers and immune scores, and are carried out on post-treatment patients [17,18,19,20]. These predictive models for NPC may have limited sensitivity in the early stages and in patients receiving treatment as well as limited specificity in the presence of other EBV-related diseases. To ensure accurate screening and prognosis of NPC, it is necessary to develop more sophisticated prediction models based on EBV-related features. In this study, we screened EBV-related host genes from NPC cell lines and conducted prognostic risk stratification in treatment-naïve NPC patients, aiming to identify core genes that can predict the prognosis of nasopharyngeal carcinoma and provide insight into the clinical prognostic assessment of NPC.

## 2. Materials and Methods

### 2.1. Microarray Dataset Collection and Preprocessing

The datasets GSE2371 [21], GSE12452 [22], and GSE102349 [23] were downloaded from the Gene Expression Omnibus (GEO) database (https://www.ncbi.nlm.nih.gov/gds/ accessed on 25 December 2022).

Dataset GSE2371 consists of five paired arrays including EBV-positive or EBV-negative representative NPC cell lines (TW01, TW03, TW04, TW06, and CGBM1). This dataset was preprocessed using the preprocessCore package in R. All genes, including housekeeping genes, were ranked and average values were calculated. Then, original values were replaced with corresponding averages and repeats for each gene using the same reference distribution. This ensures fair comparisons and removes biases, allowing for accurate analysis of gene expression levels. The differentially expressed genes (DEGs) between the EBV-positive and EBV-negative NPC cells were identified with the Limma package. The thresholds for differential genes were set at *p* < 0.05.

Dataset GSE12452 includes 41 arrays, which were collected from 31 NPC tissue samples from 31 different patients and 10 normal nasopharyngeal tissues. In the GSE102349 dataset, there are 113 tumor samples obtained from NPC patients with a 27.3-month median follow-up time. The NPC clinical specimens in these two sets of data are all EBV-positive, as evidenced by the detection of EBV-related RNA/genes [22,23]. These two clinical datasets were debatched using the combat function in the sva package into an integrated set containing 154 samples. Principal component analysis (PCA) (drawn using the FactoMineR and factoextr packages) was applied to visualize the merged datasets. The combined datasets were used to identify prognosis-associated genes.

### 2.2. Functional Enrichment Analysis

Gene Ontology (GO) and Kyoto Encyclopedia of Genes and Genomes (KEGG) enrichment analyses were performed using the ClusterProfiler package to explore the biological functions of the DEGs. The *p*-value was adjusted using the Benjamini–Hochberg approach or the false discovery rate (FDR) for multiple testing corrections. The threshold was set at FDR < 0.05. GO categories comprised biological processes (BP), molecular functions (MF), and cellular components (CC). Scatter diagrams were used to visualize the analysis results.

### 2.3. Univariate Cox Regression and Survival Analysis

Univariate Cox regression was then used to identify the DEGs correlated with prognosis. The R packages “survival” and “survminer” were used to carry out the Kaplan–Meier survival analysis.

### 2.4. Unsupervised Hierarchical Clustering

The ConsensusClusterPlus package was performed on the 25 EBV-associated genes (differential expression genes between EBV− and EBV+ NPC cells; the analysis did not include EBV-encoded genes due to insufficient information) to classify the NPC patients. An agglomerative km clustering algorithm was used, and the Pearson’s correlation distance was calculated. The optimal number of clusters was determined using empirical cumulative distribution function plots.

### 2.5. Gene Set Variation Analysis (GSVA) in EBV-Associated NPC Subtypes

The R package GSVA was used to analyze the differences in the pathways between the two EBV-associated subtypes. KEGG, BIOCARTA, and REACTOME gene sets were downloaded from MSigDB. The pathways enriched in the two subtypes were visualized using heatmaps that were generated by the pheatmap package.

### 2.6. Immune Infiltration Analysis in EBV-Associated NPC Subtypes

ESTIMATE and ssGSEA were used in R to evaluate the immune infiltration status of each EBV-associated NPC subtype. ESTIMATE was applied to assess the stromal score, immune score, and the sum of the two scores. Immune infiltration scores of 23 tumor-infiltrating immune cells were quantified using ssGSEA.

### 2.7. Supervised Learning Approaches to Identify EBV-Associated Hub Genes

Two machine learning methods were implemented to identify EBV-associated hub genes. First, the randomForest package in R was used to rank the importance of EBV-associated prognostic genes and EBV-associated NPC occurrence genes using the random forest method, which is an ensemble method of decision tree algorithms. The method involves randomly sampling features from the dataset to build multiple different trees that are combined using a voting approach to give an overall result. Second, the kernlab package in R was used to implement Support Vector Machine (SVM) and generate linear models. SVM is a supervised machine learning algorithm that uses the best hyperplane to classify data in a high-dimensional space.

### 2.8. Correlation Analysis

Pearson correlation analysis was performed between six hub genes. After calculating the correlation coefficient using the corrplot package, a Circos plot of the correlation network was visualized using the circlize package in R.

### 2.9. Prediction of Upstream Regulatory Networks of Hub Genes

The Regnetwork database (https://regnetworkweb.org/, acceseed on 25 December 2022) was used to predict upstream microRNAs (miRNAs) and transcription factors of the genes. Cytoscape software was used for visualization of the predicted upstream regulatory networks.

### 2.10. Statistical Analysis

All statistical analyses were conducted using R software 4.2.2. The differences between groups were analyzed using a Wilcoxon or Student’s *t*-test. ROC curve analysis was used to predict the NPC survival status and occurrence. A Pearson’s or Spearman’s correlation test was used to determine the correlation between genes. A two-sided *p*-value < 0.05 was considered statistically significant.

## 3. Results

### 3.1. Identification of Genes Associated with EBV Infection in NPC Cells

The GSE2371 dataset comprises five EBV-positive and five EBV-negative cell lines. To identify the differentially expressed genes (DEGs) associated with EBV (EBV-DEGs), the raw expression data underwent quantile normalization to obtain standardized microarray data (Figure 1A, without normalization; Figure 1B, normalized). After preprocessing the raw data, we performed a differential analysis using the limma package. The analysis identified 366 EBV-DEGs, determined using a threshold *p*-value < 0.05. The top 20 most significant EBV-DEGs are displayed using a heatmap (Figure 1C). To gain a deeper understanding of the biological pathways and processes associated with EBV-DEGs, we performed GO and KEGG enrichment using the R package “clusterprofiler”. The results indicated that EBV-DEGs were mainly involved in active cell membrane dynamics and pathways associated with viral infection (Figure 1C, GO term enrichment; Figure 1D, KEGG pathway enrichment). We observed an inconsistency in the gene expression profiles between different microarrays within the EBV+ group, possibly due to sample variations. However, there is still a significant discrepancy between the EBV− and EBV+ groups.

### 3.2. Screening of Prognosis-Associated Genes among the EBV-DEGs

To determine whether the EBV-DEGs identified from the cell lines have clinical significance, further analyses were conducted to evaluate their prognostic risk level and expression levels in clinical samples. The raw datasets GSE12452 (including 10 normal nasopharyngeal tissues and 31 NPC tissues) and GSE102349 (including 113 NPC patients with progression-free survival) were merged, and the combined expression profile was visualized by PCA (Figure 2A). After removing the batch effect between the two datasets, 18,295 genes in 154 clinical samples were obtained (Figure 2B). A univariate Cox regression analysis was performed on the combined expression profiles of 366 EBV-DEGs. One hundred and nine genes were not found in the merged dataset. We obtained 25 prognosis-associated EBV-DEGs from the remaining 257 genes for further analysis (Appendix A; Cox regression *p*-value: 0.01). Figure 2C shows the 11 favorable factors (PDE1A, ABCB1, NEDD8, ARF6, KCNJ8, P2RY2, ECM2, P2RX1, RAMP3, CTSW, and TMSB4X) and 14 risk factors (CYP26A1, SRPX, BCHE, GNAZ, CRMP1, ADCY2, MAPK8IP2, ENO2, MIF, REPS2, TRPS1, CEBPG, SLC22A1, and BAI3) in NPC. Significant correlations between them are also displayed (all correlations with *p* > 0.0001 were filtered out). The Kaplan–Meier (KM) survival analysis revealed that the progression-free survival (PFS) was statistically significant in those patients with differentially expressed favorable or risk factors (Figure 3, *p* < 0.05). These results indicate that the 25 screened EBV-DEGs have clinical significance in NPC patients.

### 3.3. NPC Prognostic Risk Stratification Based on the EBV-DEGs

To better understand the EBV-associated signature in NPC patients, unsupervised clustering was performed on the 25 identified EBV-DEGs. This had optimal dissimilarity when classifying all NPC samples into two clusters (Figure 4A). Subsequent survival analysis between the two clusters revealed a significant difference in prognosis, with cluster A associated with a low prognostic risk and cluster B with a high prognostic risk (Figure 4B, *p* = 0.012). In these two clusters, the majority (20/25, 80%) of the EBV-associated DEGs showed a significant difference in gene expression (Figure 4C, *p* < 0.05). The corresponding gene expression level, survival outcomes, and full survival times are further displayed using a heatmap (Figure 4D). These results suggest that NPC patients can be effectively stratified into high and low prognostic risk groups based on the 25 EBV-DEGs.

### 3.4. Gene Set Variation Analysis and Immune Infiltration Differences in EBV-Associated NPC Clusters

To provide more insight into the underlying biology in EBV-associated NPC clusters, we downloaded HALLMARK, KEGG, and Reactome pathway datasets from MSigDB (https://www.gsea-msigdb.org/gsea/msigdb/index.jsp, accessed on 10 January 2023) and scored the pathway items using the R package “GSVA”. The pathway scores between the two clusters were compared and visualized using the R package “pheatmap” (Figure 5A, HALLMARK pathway enrichment; Figure 5B, KEGG pathway enrichment; Figure 5C, Reactome pathway enrichment). The enrichment of immune regulation pathways in cluster A suggests that the immune system may be abnormally activated.

To evaluate the tumor immune microenvironments in these two clusters, we used the R package “ESTIMATE” to calculate immune scores, stromal scores, and their sum in different subtypes (Figure 6B). The infiltration scores of 23 immune cells were also calculated and compared (Figure 6C). Consistent with the results of GSVA, cluster A exhibited extensive immune cell infiltration, indicating the potential for a better prognosis and favorable response to immunotherapy.

### 3.5. Overview of Gene Expression and Pathway Enrichment in EBV-Associated NPC Clusters

To further decipher the gene expression characteristics of the EBV-associated prognosis signature in NPC patients, we performed a gene differential expression analysis on cluster A and B patients and obtained 292 DEGs with log[fold change (FC)] > 1 and *p* < 0.05 as thresholds (Figure 7A). We considered these genes to be EBV-associated subtype DEGs and named them EBV-typeDEGs. Subsequent GO enrichment analysis revealed that the EBV-typeDEGs are involved in immune system function regulation, cell membrane receptors, and signal transduction (Figure 7B; adjusted *p* < 0.05; top 10 items presented in BP, CC, and MF). The KEGG analysis showed enrichment of 29 pathways (adjusted *p* < 0.05), including “Epstein–Barr virus infection” (adjusted *p* = 0.03, Appendix A). The top 20 enriched pathways are shown in Figure 7C. The top five enriched pathways include primary immunodeficiency, cytokine–cytokine receptor interaction, hematopoietic cell lineage, viral protein interaction with cytokine and cytokine receptor, and the B cell receptor signaling pathway, which indicates that the corresponding EBV-typeDEGs have potential connections with functional deficits of the immune system and the interactions between viral proteins, cytokines, and cytokine receptors (Figure 7D).

### 3.6. Selecting Hub Genes from the EBV-typeDEGs

To identify key EBV-associated genes related to the prognosis of NPC, a single-factor regression analysis was performed on the 292 EBV-typeDEGs, and 48 prognostic genes were screened when *p* < 0.005 (Figure 8A, Appendix A), which were named EBV-proDEGs. A gene expression differential analysis was also performed on the GSE12452 and GSE102349 merged datasets (Figure 8B,C). There were 3337 upregulated genes and 2501 downregulated genes in the NP and NPC (Figure 8D,E). By taking the intersection, although there was no intersection between the upregulated genes and 48 EBV-proDEGs (Figure 8F), we obtained 11 downregulated genes in tumors that are associated with prognosis (Figure 8G,H) for later analysis.

To further identify key EBV-associated genes involved in the occurrence and development of EBV, machine learning algorithms, including support vector machine (SVM) and random forest, were performed on the 11 selected EBV-proDEGs (Figure 9A–D). This resulted in ten NPC prognostic-related genes and six NPC occurrence-related genes, with an intersection of six genes being identified as an EBV-associated occurrence and prognosis gene signature in NPC (C16orf54, CD27, CD53, CRIP1, RARRES3, and TBC1D10C; Figure 9E). A correlation analysis between the six genes showed a strong mutual correlation (Figure 9F). The receiver operating characteristic (ROC) curves of the six hub genes were analyzed to evaluate their ability to predict the occurrence and progression of NPC, with all areas under the curve (AUCs) above 0.6. Taking the genes individually, CRIP1 showed the best predictive performance (AUC of survival = 0.741; AUC of occurrence = 0.851) (Figure 9G, survival; Figure 9H, occurrence). These results indicate that the six selected hub genes have prognostic value in NPC.

### 3.7. The Relationship between Hub Genes with Immune Infiltration and Gene Set Enrichment Analysis (GSEA)

Subsequently, we conducted an analysis of the infiltration proportions of 23 immune cells and found that, except for CD56dim NK cells and T helper type 17 (Th17) cells, the infiltration of the remaining cells showed a good correlation (Figure 10A). Fourteen of the 23 immune cells showed significant differences between normal and NPC patients (Figure 10B). We also analyzed the correlation between the six hub genes and immune cell infiltration. All six genes were positively correlated with most immune cells. Interestingly, T helper type 2 (Th2) cells, which may secrete Th2 cytokines and are involved in immune escape [24], showed a negative correlation with these genes (Figure 10C).

To investigate the genes and signaling pathways associated with the six hub genes, we conducted a correlation analysis and a pathway enrichment analysis. The top 50 positively and negatively correlated genes are shown in Figure 11. Additionally, the correlations between each hub gene and the enriched pathways are illustrated in Figure 12, which indicate that the downregulated hub genes C16orf54, CD27, CD53, CRIP1, RARRES3, and TBC1D10C are highly enriched in immune-related pathways while showing a negative correlation with cell cycle-related signaling pathways.

### 3.8. Prediction of Upstream miRNAs and Transcription Factors of the Hub Genes

Finally, we used the Regnetwork database (https://regnetworkweb.org/, accessed on 10 January 2023) to predict the miRNAs and transcription factors upstream of C16orf54, CD27, CD53, CRIP1, RARRES3, and TBC1D10C (Figure 13). Although no upstream factors were predicted for CD27, the results identified a subset of miRNAs and transcription factors that are associated with tumor proliferation, invasion, and immune regulation.

## 4. Discussion

Through the novel integration of the effects of EBV infection in cell lines and the survival status in treatment-naïve NPC patients, this study identified six hub genes that can predict the prognosis of NPC based on a series of sequential analyses together with machine learning algorithms. This prediction model may provide insight into the mechanisms of development as well as the clinical treatment of NPC.

EBV is almost always present in NPC [25], making it difficult to identify the EBV-specific host genes from EBV-negative NPC tissues. To address this issue, we screened for EBV-associated DEGs from EBV-negative and EBV-positive cell pairs and conducted a single-factor regression analysis on the gene expression profiles of NPC patients. As a result, we identified 25 DEGs that are both related to EBV and have prognostic significance.

Based on these DEGs, we classified NPC patients into two subtypes: cluster A and cluster B. A pathway enrichment and immune infiltration analysis showed significant differences between the two types, indicating that the 25 DEGs effectively distinguished NPC patients. Notably, the immune regulation pathways of cluster A were highly activated and infiltrated with a large number of immune cells, while cluster B was the opposite. It has previously been concluded that EBV-associated NPC, which is characterized by dense infiltration of immune cells, is a typical type of “immune-hot” tumor [7]. Recent studies have shown that NPC may exert an “immune-cold” or immune-quiescent phenotype in which the immune infiltration is extremely limited and that patients with this type may have a poor response to immunotherapy, leading to a worse prognosis [26,27]. Our research supports this conclusion, and based on this finding, we incorporated the factor of EBV infection when stratifying the NPC patients, thus broadening the scope of research in this field.

We identified six hub genes (C16orf54, CD27, CD53, CRIP1, RARRES3, and TBC1D10C) that may predict the prognosis of NPC. CRIP1 shows the best predictive performance for both occurrence (AUC = 0.851) and survival (AUC = 0.741) of NPC. CRIP1 belongs to the LIM protein family, characterized by a double zinc finger motif, and exhibits high expression in immune cells and the intestine [28]. Prior research on CRIP1 has predominantly focused on tumors, highlighting its tumor type-specific oncogenic or tumor-suppressive characteristics [29]. CRIP1 is involved in carnitine metabolism and cancer stem-like properties in hepatocellular carcinoma [30]; DNA methylation in prostate cancer [31]; and cell migration, invasion, and the epithelial–mesenchymal transition in cervical cancer [32]. In breast cancer, patients with low CRIP1 expression have a poor prognosis [33], while in gastric cancer, high CRIP1 expression is an independent predictor of adverse prognosis [34]. In this study, we clarified that CRIP1 has a low expression in NPC, which can predict tumor occurrence and patient survival. Though the functions of C16orf54, RARRES3, and TBC1D10C in cancer development have also been previously studied [35,36,37], their roles in NPC are revealed for the first time in this study. CD27 and CD53 are often expressed in the immune system, where they play important regulatory roles. For example, CD27 is a crucial T-cell receptor that provides costimulatory signals required for optimal T-cell priming and memory differentiation. Its role in CD8+ T-cell activation, especially the cytotoxic pathway, is central to the immune response and has potential applications in antitumor therapy [8,38], while CD53 plays a role in regulating immune cell signaling and participating in the adhesion and migration of immune cells [39].

In addition, the relationship between the hub genes suggests that they are involved in immune-related and cell cycle-related signaling pathways. Furthermore, we predict that these hub genes may be regulated by various miRNAs and transcription factors, which may provide new insight into the regulatory mechanisms of NPC development and progression. Nevertheless, further experimental investigations are necessary to validate our results. Moreover, to confirm our bioinformatics findings, more independent datasets should also be included in future studies.

Despite the challenges associated with targeting downregulated genes in cancer cells, Feng Z et al. proposed a promising approach for addressing this issue. By combining enzyme-instructed assembly and disassembly, they were able to target downregulation in cancer cells using peptidic precursors designed to be substrates for both carboxylesterases (CESs) and alkaline phosphatases (ALPs) [40]. This approach shows great potential for developing new therapies for cancer treatment.

In this study, we focused on analyzing the expression profiles of EBV-associated genes in NPC tissue samples using bulk RNA sequencing data. However, single-cell RNA sequencing provides a more detailed and comprehensive understanding of the cellular landscape in NPC. Recent single-cell RNA sequencing analyses provided evidence related to microenvironment interactions, immune cell diversity/subtypes, and tumor heterogeneity in NPC [41,42,43,44], so comparing our study with single-cell RNA sequencing data is necessary in the future.

Our research is also a good inspiration for future studies. For example, research can be performed to explore or confirm the biological role of these hub genes as well as to explore the clinical applicability of these genes in patients and to evaluate patients’ prognosis or treatment outcomes.

## 5. Conclusions

In conclusion, our study provides valuable insight into the clinical prognostic assessment of NPC and suggests that the six hub genes identified may serve as potential biomarkers for predicting the prognosis of NPC based on EBV-related features.

## Figures and Tables

**Figure 1 viruses-15-01915-f001:**
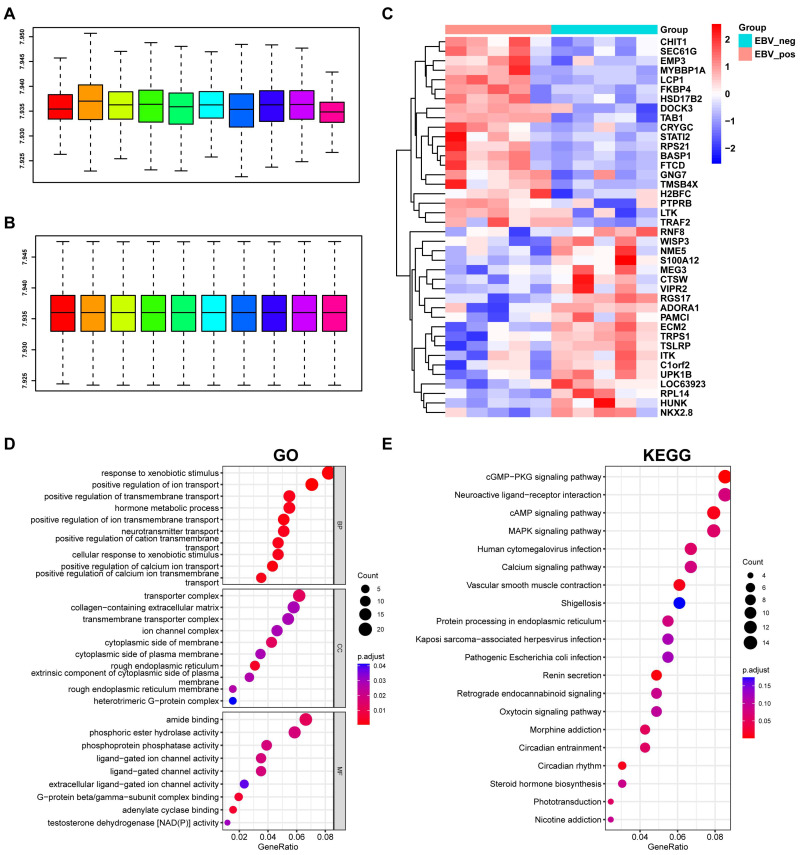
Identification of genes associated with EBV infection in the GSE2371 dataset. (**A**,**B**) The gene expression profile of the raw data from GSE2371 before and after normalization. The bars represent all samples included in the analysis, with each color indicating a different sample. (**C**) Heatmap of the representative 20 up- and 20 downregulated EBV-associated genes. Pink and cyan indicate EBV-positive and EBV-negative NPC cell lines, respectively. Red and blue indicate up- and downregulated EBV-DEGs, respectively (*p* < 0.05). (**D**) GO term enrichment of the EBV-DEGs. BP, biological process; CC, cellular components; MF, molecular function. (**E**) KEGG pathway enrichment of the EBV-DEGs.

**Figure 2 viruses-15-01915-f002:**
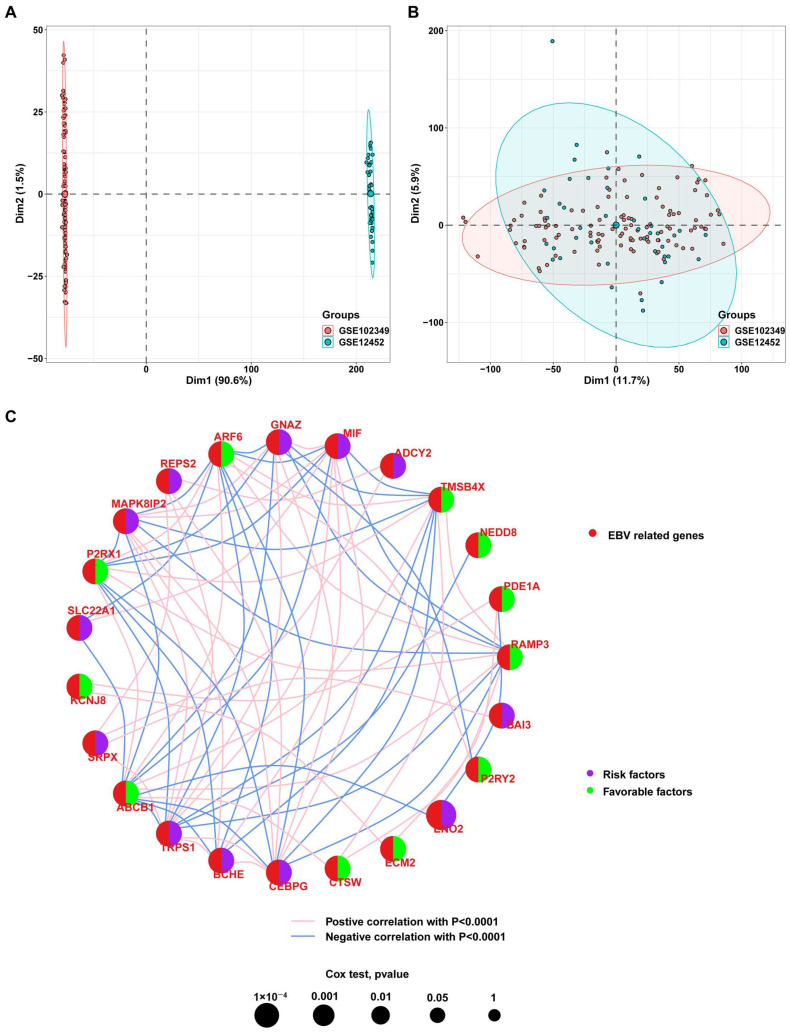
Screening of prognosis-associated genes in the combined datasets of GSE12452 and GSE102349. (**A**,**B**) Principal component analysis (PCA) of the combined datasets ((**A**) without removing batch effects; (**B**) removing batch effects). (**C**) Correlation network of 25 prognosis-related EBV-DEGs. Purple and cyan dots indicate risk and favorable factors, respectively. Pink and blue lines indicate positive and negative correlations, respectively, between two genes (*p* < 0.0001).

**Figure 3 viruses-15-01915-f003:**
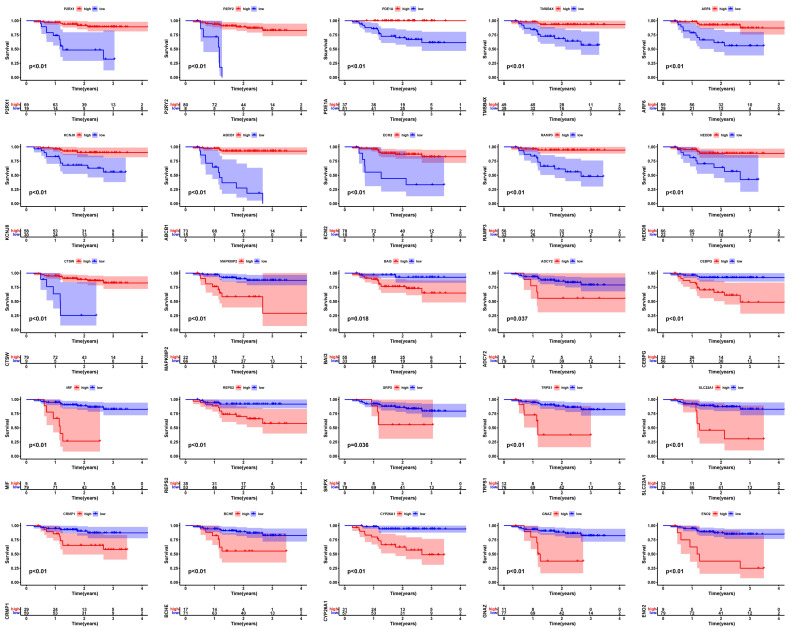
Kaplan–Meier (KM) survival curves of 25 prognosis-associated EBV-DEGs.

**Figure 4 viruses-15-01915-f004:**
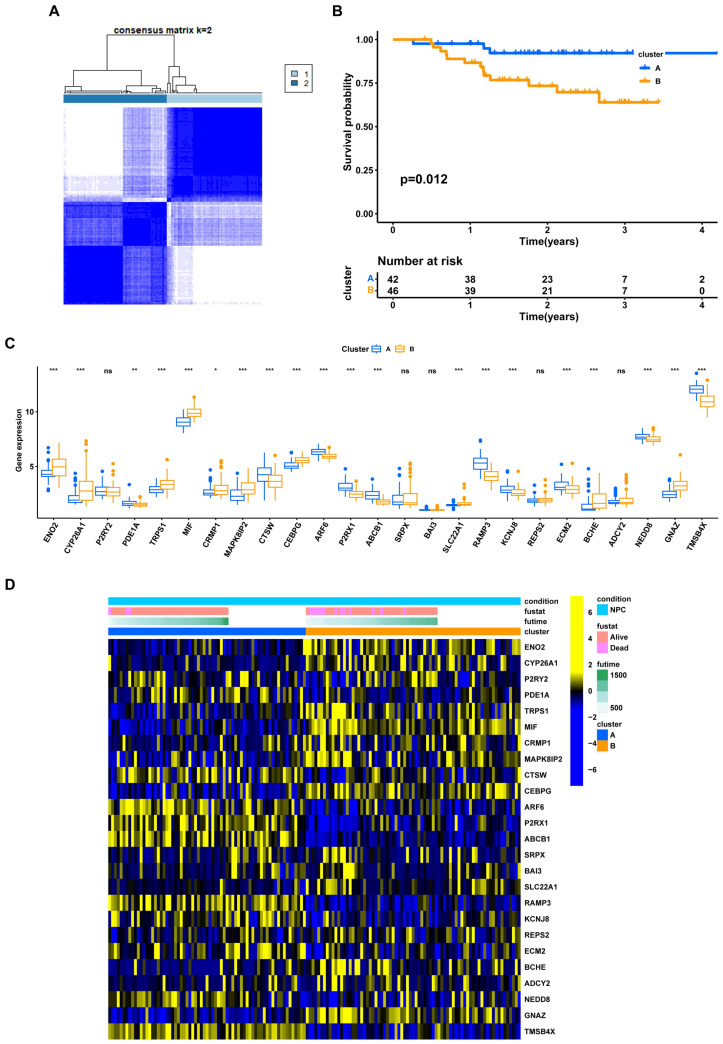
Unsupervised clustering of 25 EBV-associated genes. (**A**) Consensus matrix heatmap. The optimal number of clusters: k = 2. (**B**) Kaplan–Meier (KM) survival curves showing the survival probability of the two EBV-associated prognosis clusters (*p* = 0.012). (**C**) The expression level of 25 prognosis-associated EBV-DEGs in the two EBV-associated prognosis clusters (*, *p* < 0.05; **, *p* < 0.01; ***, *p* < 0.001; ns, no significance). (**D**) Heatmap showing the expression level of 25 prognosis-associated EBV-DEGs, the survival outcomes (alive or dead), and full survival times (days) in the two EBV-associated prognosis clusters.

**Figure 5 viruses-15-01915-f005:**
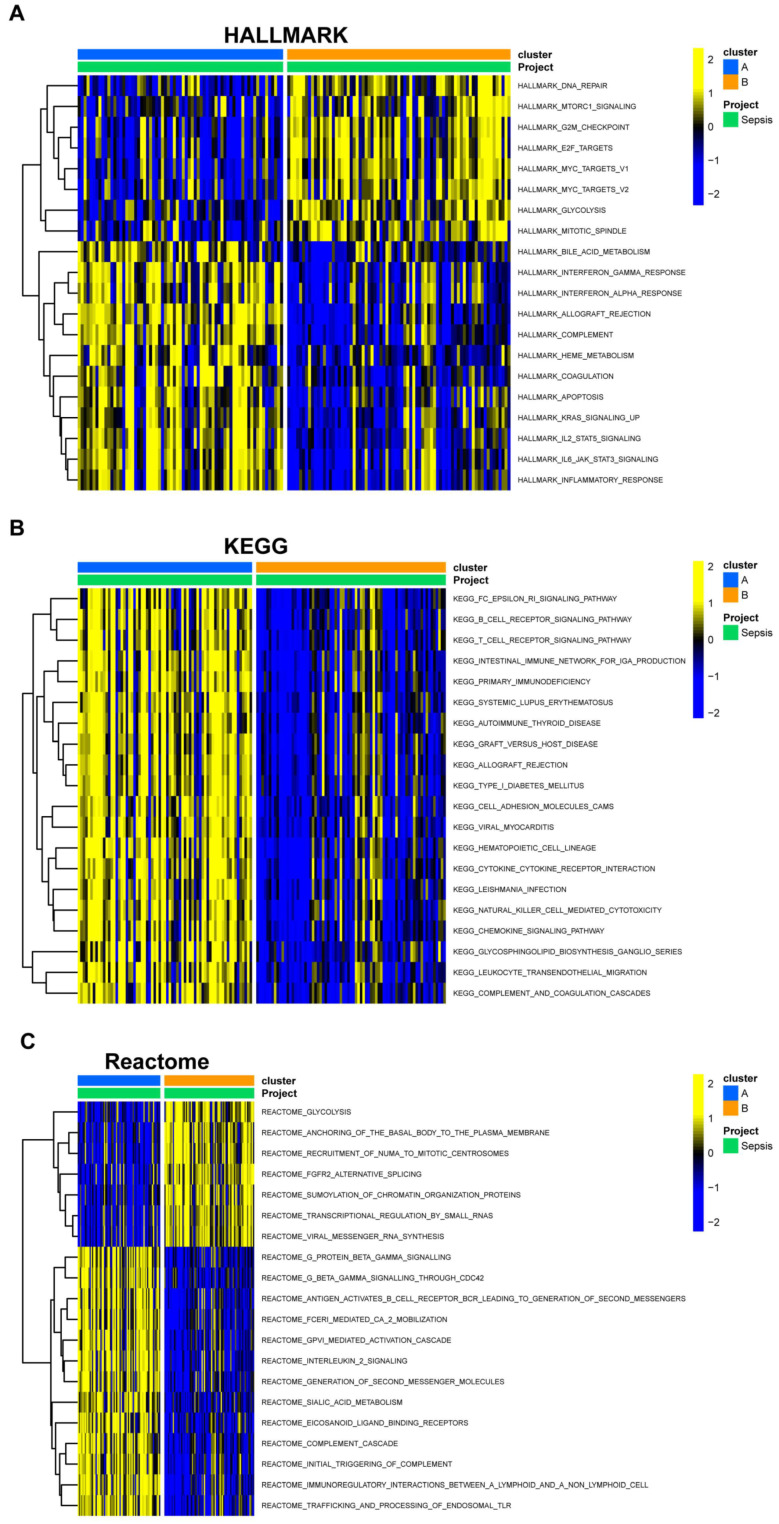
Pathway enrichment analysis in the two EBV-associated prognosis clusters. (**A**) HALLMARK, (**B**) KEGG, and (**C**) Reactome pathway differences in NPC prognosis cluster A and B. Blue, cluster A; yellow, cluster B.

**Figure 6 viruses-15-01915-f006:**
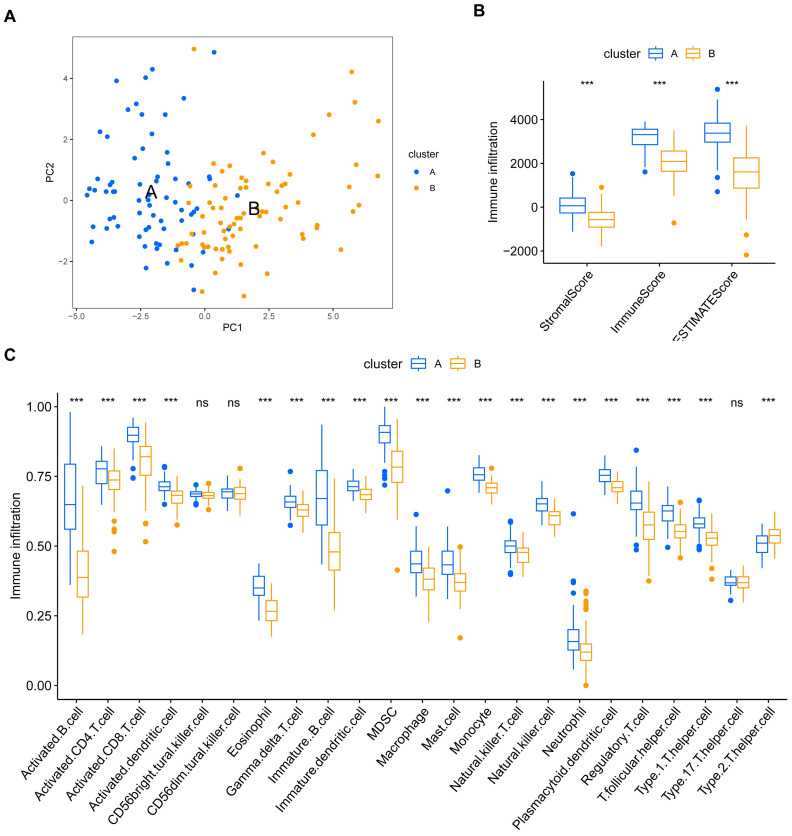
Immune infiltration analysis in the two EBV-associated prognosis clusters. (**A**) Principal component analysis (PCA) results are visualized using scatter dot plots, which depict the samples in the two clusters. (**B**) Differences in StromalScore, ImmuneScore, and ESTIMATEScore in the two clusters. (**C**) Differences in 23 types of immune-infiltrating cell enrichment in the two clusters. Blue, cluster A; yellow, cluster B; ***, *p* < 0.001; ns, no significance.

**Figure 7 viruses-15-01915-f007:**
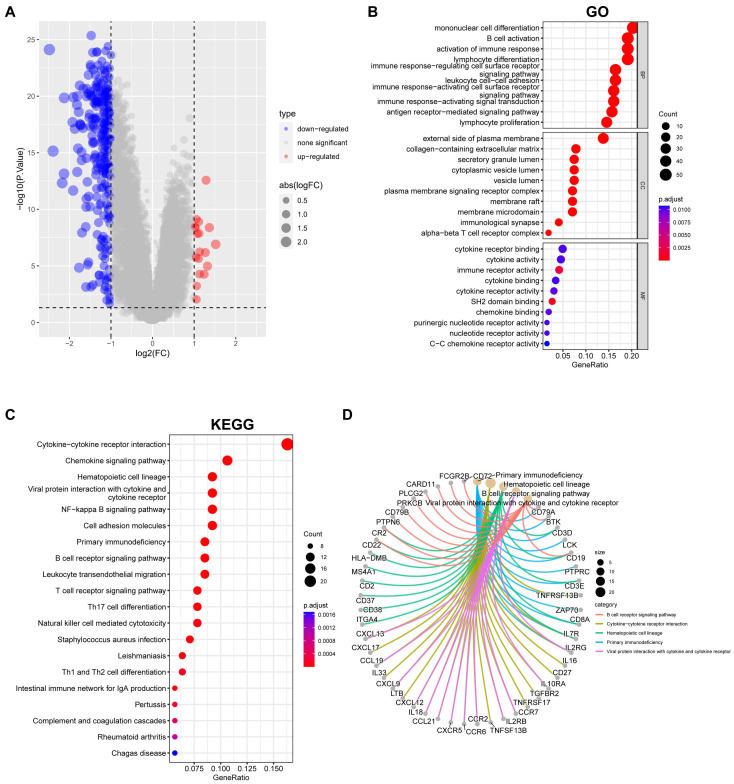
Overview of gene expression and pathway enrichment in EBV-associated low-/high-risk NPC patients. (**A**) Volcano plots of DEGs between cluster A and cluster B. The threshold is set at logFC > 1 and *p* < 0.05. (**B**) GO enrichment items. (**C**) KEGG enrichment items. (**D**) The correspondence between the top five KEGG enriched items and the DEGs.

**Figure 8 viruses-15-01915-f008:**
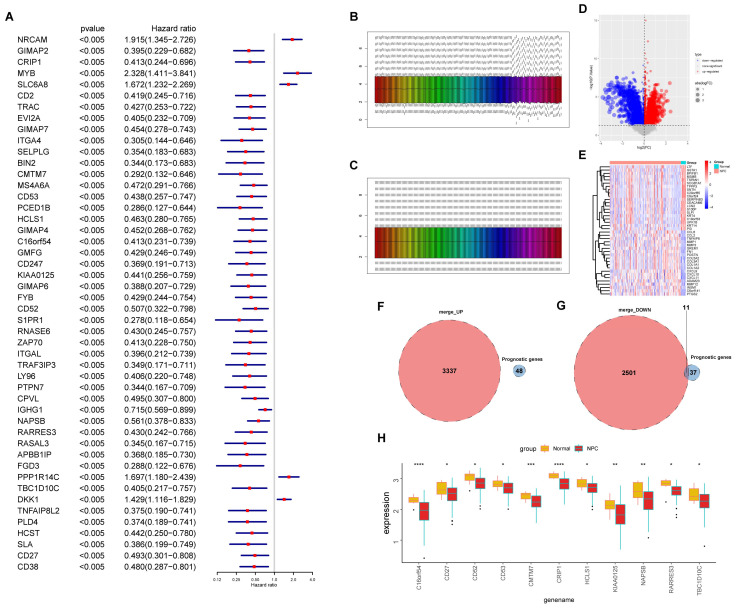
Identification of the DEGs between NP and NPC based on DEGs in EBV-associated low-/high-risk NPC patients. (**A**) Forest plot of 48 prognosis-associated DEGs in NPC when the significance is set at *p* < 0.005. (**B**,**C**) Gene expression levels in the merged dataset without or after debatching. The bars represent all samples included in the analysis, with each color indicating a different sample. (**D**) Volcano plots showing DEGs between NP and NPC samples in the merged dataset. (**E**) Heatmap showing the top 20 up- and downregulated genes. (**F**) Venn diagram of 3337 upregulated genes and 48 prognosis-associated DEGs. (**G**) Venn diagram of 2501 downregulated genes and 48 prognosis-associated DEGs. (**H**) Box plot of 11 downregulated genes. *, *p* < 0.05; **, *p* < 0.01; ***, *p* < 0.001; ****, *p* < 0.0001; ns, no significance.

**Figure 9 viruses-15-01915-f009:**
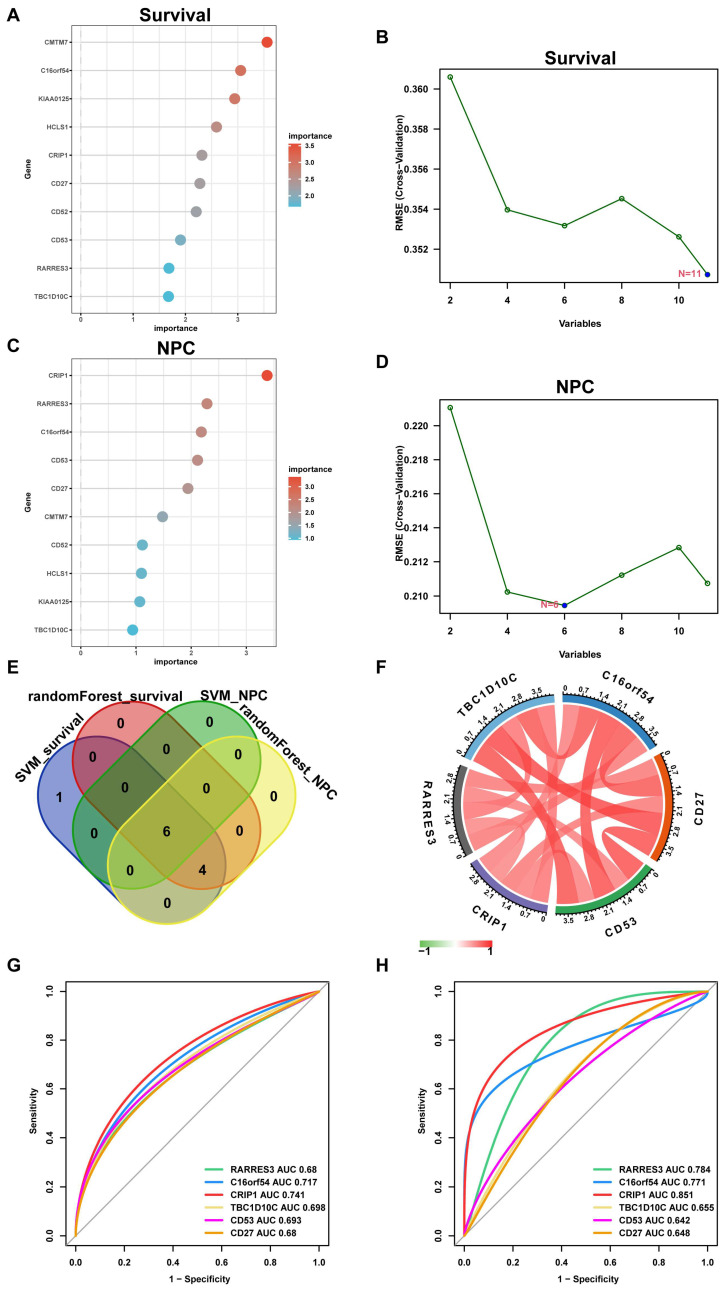
Identification of hub genes using machine learning. (**A**) Construction of prognosis-associated candidate genes by random forest. (**B**) Screening of prognosis-associated candidate genes by SVM. (**C**) Construction of an occurrence-associated candidate gene signature by random forest. (**D**) Screening of occurrence-associated candidate genes by SVM. (**E**) Venn diagram showing the overlap of the candidate genes in (**A**–**D**). (**F**) Circos plot displaying the relationship between the overlapping genes (hub genes) in E. (**G**) ROC curve of the NPC prognosis signature. (**H**) ROC curve of the NPC occurrence signature.

**Figure 10 viruses-15-01915-f010:**
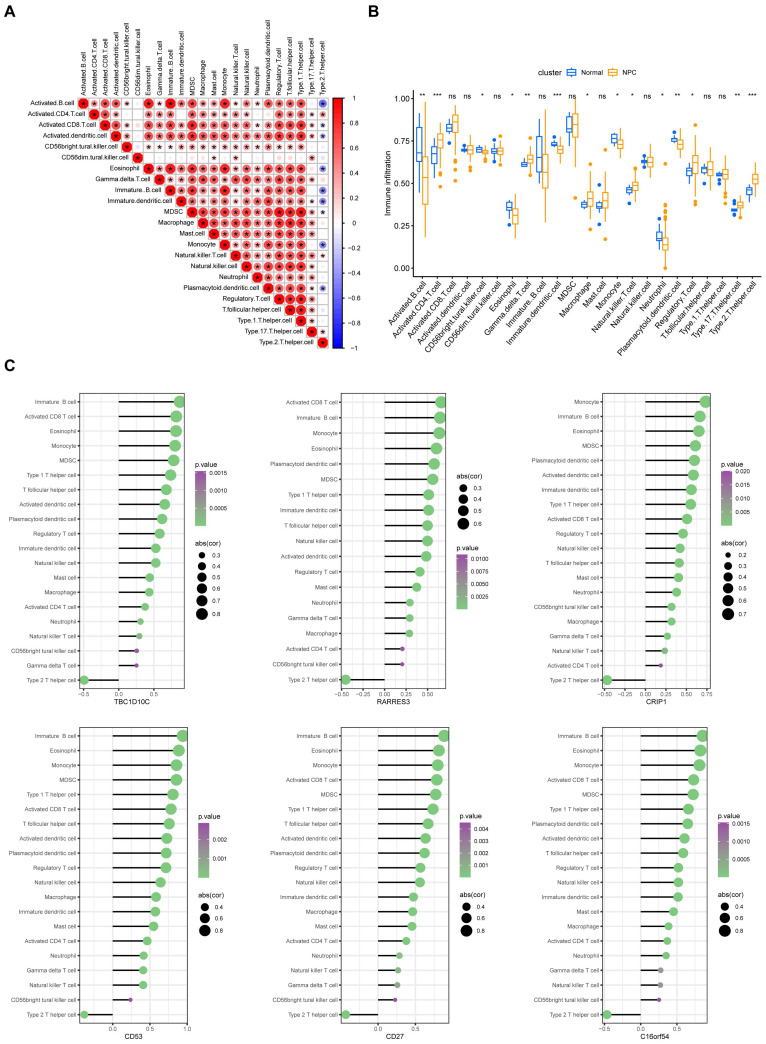
Hub gene-related immune infiltration analysis. (**A**) Comparison of 23 immune cell subtypes between NC and NPC patients. (**B**) Correlation matrix of all 23 immune cell subtype compositions in NC and NPC patients. (**C**) Correlation between hub genes and immune-infiltrating cells. *, *p* < 0.05; **, *p* < 0.01; ***, *p* < 0.001; ns, no significance.

**Figure 11 viruses-15-01915-f011:**
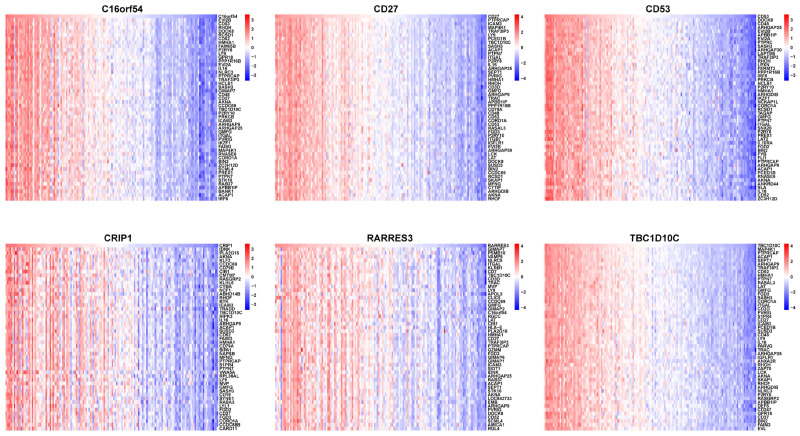
Correlation analysis between the six hub genes and all genes expressed in NPC. The top 50 positive and negative correlation genes are displayed in the heatmaps.

**Figure 12 viruses-15-01915-f012:**
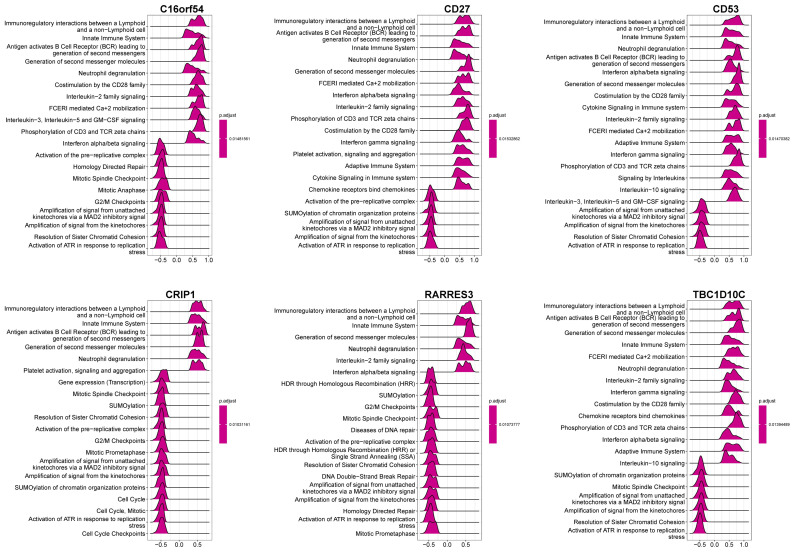
GSEA of the six hub genes. The top 20 items are displayed in the enrichment map.

**Figure 13 viruses-15-01915-f013:**
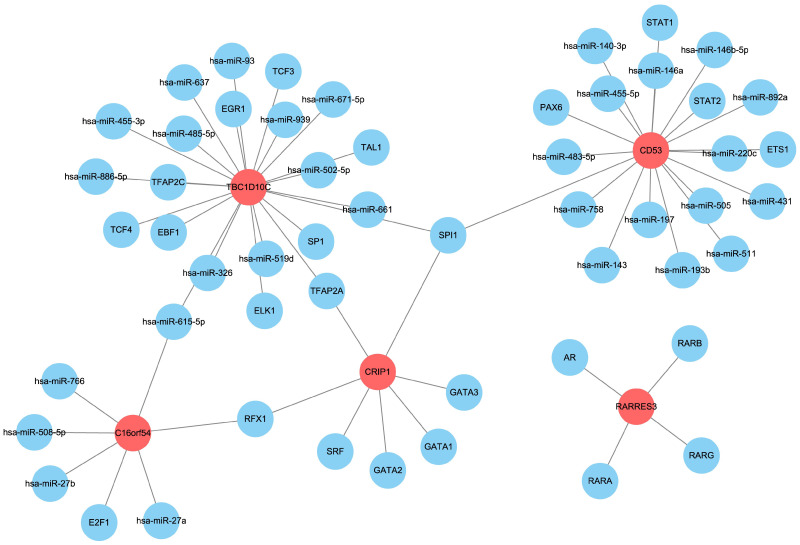
The miRNA and transcription factor regulatory network of the six hub genes.

## Data Availability

The data that supported the findings of this study are openly available in the Gene Expression Omnibus (GEO) database: https://www.ncbi.nlm.nih.gov/geo/. Data generated from analysis in this study are available from the corresponding author upon reasonable request.

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
