# Peer review of "EBV-Associated Hub Genes as Potential Biomarkers for Predicting the Prognosis of Nasopharyngeal Carcinoma"

_viruses, 2023, doi:10.3390/v15091915_

Round 1

Reviewer 1 Report

In this work by Tengteng Ding et al. Titled “EBV-Associated Host Genesas Potential Biomarkers for Predicting the Occurrence and Prognosis of Nasopharyngeal Carcinoma”, investigated the potential presence of Biomarkers by comparing genes differentially expressed in three different already published datasets. The authors Used three datasets GSE2371, GSE12452 and GSE102349 that were composed of respectively of: EBV+ and EBV- cell lines, 31 Nasopharyngeal cancer tissue coupled with 10 normal Nasopharyngeal tissues and 115 Tumour samples which had up to a mean of 27 months follow up. By comparing the Raw data of this datasets, the Authors analysed which  host  genes associated with EBV were more probably involved  prognosis and their biological function in different mechanisms and settings.

 They identified six hub genes (C16orf54, CD27, CD53, CRIP1, RARRES3, and 354 TBC1D10C) which may predict the occurrence and prognosis of NPC.

Author Response

Thank you so much for your valuable comments and feedback on our manuscript. We highly appreciate your time and effort in reviewing our work.

We appreciate your recognition of the significance of our findings regarding the potential biomarkers for NPC and their clinical implications. We also totally agree with your comments. By utilizing 3 datasets and conducting a comprehensive analysis, we were able to identify six core genes that could play a crucial role in predicting the prognosis of NPC. These findings provide new ideas for prognosis evaluation and research on new targeted therapeutic strategies for NPC patients.  

Reviewer 2 Report

In T. Ding et al.’s study, a new model is developed to predict the occurrence and prognosis of nasopharyngeal carcinoma (NPC) using Epstein-Barr virus (EBV)-associated host genes. This is an interesting and scientifically valuable study. However, I have several concerns about the data analysis in the study.

1.     How to define the “EBV-associated genes”. Are all the “EBV-associated genes” encoded by the EBV genome? Or functionally related genes? Please clearly define.

2.     It has been well established that EBV-encoded genes, like LMP1 and LMP2A, are functionally associated with malignant biological behavior and clinical prognosis. However, such EBV-encoded genes are not present in the present study. The authors should explain and discuss this issue.

3.     In recent years, single-cell RNA-seq has revealed the landscape of tumor and infiltrating immune cells in NPC (Cell Res. 2020;30(11):950-965.; Cell Res. 2020;30(11):1024-1042; Nat Commun. 2021;12(1):741.; Nat Commun. 2021;12(1):1540.). The authors should compare their results with the above studies.

Reviewer 3 Report

Ding et al. present an in-silico analysis of multiple NPC microarray data sets and suggest the identification of six hub genes that their expression profile could be used for the prediction of occurrence and prognosis of NPC.

I have the following points:

My major concern with the analysis is that the authors used three data sets:

1.     GSE2371 consists of five pairs of microarrays of EBV-positive or EBV-negative representative NPC cell lines

2.     GSE12452 includes 41 arrays, which were collected from 31 NPC tissue samples from 31 different patients and 10 normal nasopharyngeal tissues. No information about EBV infection

3.     GSE102349 includes 113 tumor samples obtained from NPC patients with a 27.3 months median follow-up time. No information about EBV infection

So the last two data sets had no information about EBV infection and the first one includes cell line.

Although it is tempting to assume that all NPC cases could be infected with EBV, this is not necessarily the case. Therefore, the study is based on the assumption that the samples in the last two data sets are EBV positive.

The authors picked up EBV-DEGs from the first data set and applied them to the other two data sets despite the fact that these data sets may contain EBV negative cases. I do not understand the need to include EBV here. The analysis could be done on all NPC cases compared to normal individuals.

Since the authors did not mention the stage of tumor in set 2 and they combined set 2 and 3, then we are not sure about the use of the identified hub genes to predict the occurrence of NPC because we do not know at what stage the genes were deregulated.

This also means that we cannot use them to predict the occurrence because they are already present when we detected NPC.

Furthermore, if we assume to accept these hub genes for prediction of occurrence and prognosis of NPC, then how do we apply them? what are the target risk groups?

Do we have first to test for EBV which infects a huge number of the population and then for these genes, which might be normal before the occurrence of cancer?

In my opinion, the analysis should be either between NPC cases and normal tissue or the three groups of EBV +ve NPC, EBV -ve NPC, and normal tissue. It is also difficult to study an upregulated gene in the state of cancer and say we will use to predict the occurrence of cancer because we do not know its state before progressing to cancer. This can only be performed in a prospective study not retrospective data analysis

Here are some other points:

Abstract. It should be clarified in the abstract that this analysis was performed on data sets available from databases

Results:

Section 3.1. on what bases were the data normalized? For example, certain housekeeping genes. Please state that clearly

Figure 1C. there is clear inconsistency with the expression profile of the genes between different microarrays from the same group in the heat map. For example, the five microarrays in NPC-EBV positive do not all show downregulation or upregulation of the same genes. How did you decide whether this gene is down or upregulated?

Line 366-368. The expression of these 3 genes is deregulated in the study which does not confirm their role in NPC. A statistical association does not confirm biological role. Therefore, the sentence here should be talking about potential role not confirmed role.

Table S1. Shows only 257 genes and the title mentions 366
